# TCRA-LLM: Token Compression Retrieval Augmented Large Language Model for Inference Cost Reduction

**Junyi Liu**🥚🔑, **Liangzhi Li**🥚🔑✉, **Tong Xiang**🥚🔑, **Bowen Wang**🍁🥚🔑,
**Yiming Qian**⭐

🥚Meetyou AI Lab, 🔑Xiamen Key Laboratory of Women's Internet Health Management,
🍁Osaka University, ⭐Agency for Science, Technology and Research (A*STAR)
{liujunyi, liliangzhi, xiangtong}@xiaoyouzi.com,
bowen.wang@is.ids.osaka-u.ac.jp, qiany@ihpc.a-star.edu.sg

## Abstract

Since ChatGPT released its API for public use, the number of applications built on top of commercial large language models (LLMs) increase exponentially. One popular usage of such models is leveraging its in-context learning ability and generating responses given user queries leveraging knowledge obtained by retrieval augmentation. One problem of deploying commercial retrieval-augmented LLMs is the cost due to the additionally retrieved context that largely increases the input token size of the LLMs. To mitigate this, we propose a token compression scheme that includes two methods: *summarization compression* and *semantic compression*. The first method applies a T5-based model that is fine-tuned by datasets generated using self-instruct containing samples with varying lengths and reduce token size by doing summarization. The second method further compresses the token size by removing words with lower impact on the semantic. In order to adequately evaluate the effectiveness of the proposed methods, we propose and utilize a dataset called Food-Recommendation DB (FRDB) focusing on food recommendation for women around pregnancy period or infants. Our summarization compression can reduce 65% of the retrieval token size with further 0.3% improvement on the accuracy; semantic compression provides a more flexible way to trade-off the token size with performance, for which we can reduce the token size by 20% with only 1.6% of accuracy drop.

## 1   Introduction

With the increase in computing power and accumulation of enormous text data, large language models (LLMs) such as ChatGPT (OpenAI, 2023b) and GPT-4 (OpenAI, 2023a) have shown impressive performance in dialogue-based question-answering (QA), allowing them to interact with users fluently. In open-domain QA where the models are engaged in casual conversations with users, LLMs exhibit astonishing performance by leveraging strong in-context learning ability. However LLMs may produce vague responses or incorrect answers in certain specialized domains, owing to the absence of relevant knowledge or a restricted scope of information acquired during the training stage, which might potentially result in untruthful answers and even cause physical damages to users (Xiang et al., 2023). For QA in such domains, retrieval-augmented generation (RAG) (Lewis et al., 2020), where the system retrieves external knowledge beforehand and then utilizes LLMs to generate answers leveraging retrieved knowledge, can greatly reduce the hallucinations generated (Shi et al., 2023; Shuster et al., 2021).

Many current commercial LLMs are black-box models, where the model architectures and the weight information are not disclosed. These LLMs own superior text comprehension abilities, yet in many cases they can only output desired answers through complicated prompt engineering. On the other hand, deploying open-source LLMs to local servers is resource-intensive, in contrast to deploying smaller models such as T5 (Raffel et al., 2020). Some commercial LLMs like GPT-3.5-turbo (OpenAI, 2023c) and GPT-4 offer access through API calls; however, these models charge users based on the size of input and output[1]. For individuals or companies looking to create their own services using LLMs through API calls, utilizing commercial ones can be resource-consuming if requests are made frequently. Therefore, it is necessary to minimize the number of input tokens while maintaining optimal performance during the API calls.

In this work, we propose a token compres-

✉Corresponding author.

---

[1]As of May 5th, 2023, GPT-4, capable of processing up to 8k tokens, charges $0.03 per thousand input tokens and $0.06 per thousand output tokens; GPT-4-32K which can process 32k tokens, charges $0.06 per thousand input tokens and $0.12 per thousand output tokens. GPT-3.5-turbo charges $0.002 per thousand tokens.

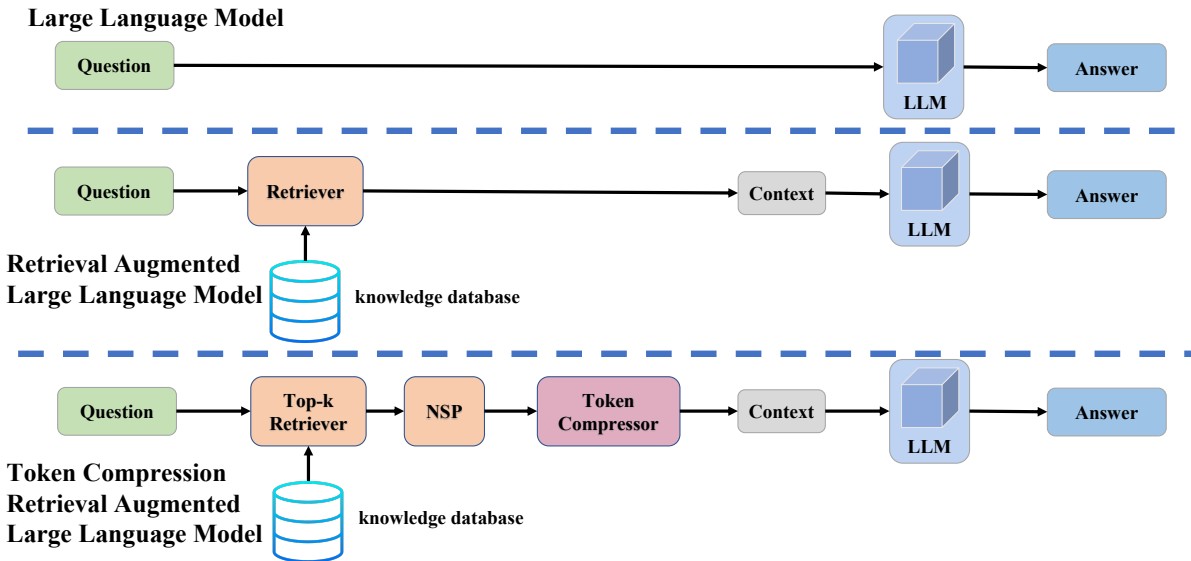

Figure 1: Illustration of different ways to utilize LLMs for QA. **Top**: directly using LLM. **Middle**: using a retrieval-augmented LLM. **Bottom**: using retrieval-augmented LLM with our proposed token compression methods.

sion scheme specifically designed for the retrieval-augmented LLMs (shown in Figure 1), namely, **T**oken **C**ompression **R**etrieval **A**ugmented **L**arge **L**anguage **M**odel (TCRA-LLM). Our proposed scheme can reduce up to 65% of the token size with additional 0.3% improvement on accuracy when doing QA on our proposed dataset called Food-Recommendation DB (FRDB). We propose two approaches to reduce the token size of the LLMs' input: *summarization compression* and *semantic compression*. For summarization compression, we leverage self-instruct (Wang et al., 2022) scheme to build multiple summarization datasets with varying lengths to fine-tune the mT5 model (Xue et al., 2020). The samples from the summarization datasets are generated by GPT-3.5-turbo (OpenAI, 2023c), which is instructed to shorten the summary of the input sentences in an iterative manner. The semantic compression approach is based on a simple yet effective intuition, that *removing semantically less important words in a sentence won't drastically change its semantic*. Here, we deploy a multi-lingual sentence-transformer (Reimers and Gurevych, 2020) to encode sentences into embeddings where the distances between original and perturbed embeddings are used to measure the semantic deviation from the original meaning. Larger semantic deviation indicates that the corresponding word owns more important semantic in the sentence. We conduct an iterative process that measures the semantic importance of each word in the sentence and remove less important words.

In conclusion, our work has the following three contributions:

1. We construct a food recommendation QA dataset (Section 3) which contains domain knowledge that general LLMs might not have. This dataset serves the purpose of evaluating retrieval-augmented LLMs.

2. We propose a multi-level self-instruct scheme (Section 4.2) to build summarization datasets of different lengths.

3. We propose two token compression methods (Section 4.2), both of which can reduce the number of input tokens during the API calls of retrieval-augmented commercial LLMs while maintaining optimal performance.

## 2 Related Work

LLMs such as GPT-3 (Brown et al., 2020), PALM (Chowdhery et al., 2022), OPT (Zhang et al., 2022), Bloom (Scao et al., 2022), and LLaMA (Touvron et al., 2023) are trained on massive amounts of data and have demonstrated powerful comprehension capabilities. These models have been deployed in a breadth of tasks and achieve promising results (Zhang et al., 2023; Ashok and Lipton, 2023; Lu et al., 2023; Wang et al., 2023; Xiang et al., 2023). One major barrier that prevents more people from participating in commercial deployment of the LLMs is

their training and hosting costs. A way to reduce such costs is through training smaller domain-specific models such as BioMedLM (Bolton et al., 2022), BloombergGPT (Wu et al., 2023), and LawGPT (Nguyen, 2023). Such domain-specific training enables smaller LLMs to be applied to certain fields but still requires huge investment. For instance, BloombergGPT is trained on 512 40GB A100 GPUs with the total budget being approximately $2.7 million (Sheikh, 2023).

Alternatively, LLMs can be used without fine-tuning through retrieval augmentation leveraging external data sources, where the retrieved data is used as supplementary information to help LLMs improve logical reasoning and language generation (Thorne et al., 2021; Izacard et al., 2022). Previous experiments (Ram et al., 2023) show that additional information can be beneficial for LLMs across different model sizes. Retrieval augmentation eliminates the cost of tuning an in-house LLM on new data, and can be easily integrated with commercial LLM services such as ChatGPT (OpenAI, 2023b) from OpenAI or Bard (Pichai, 2023) from Google. Many studies have shown, applying retrieval augmentation to the commercial LLMs such as ChatGPT allow the models to gain knowledge in specific domains such as natural science and medicine (Soong et al., 2023; Inaba et al., 2023) which is not revealed during their training and retrieval augmentation can be further improved by applying more sophisticated retrievers (Shi et al., 2023). However, commercial LLMs all have a limitation on input lengths which put an upper ceiling on the amount of information that can be fed into a LLM. Later models such as GPT-4 has looser restriction but the inference cost increases drastically in comparison with other models. Some previous work applies template-based prompt optimization (Santra et al., 2023), which select retrieved context (Mallen et al., 2022) in an adaptive manner, or uses cascading of LLMs with different sizes (Chen et al., 2023) to reduce the inference costs. Our proposed method has no conflict with these methods and can be used with them simultaneously.

## 3 FRDB

We build a Food Recommendation Dataset in Chinese called FRDB, for recommending foods that are safe to consume for women before/during/after their pregnancy as well as infants. It contains two

| Food type | Count↓ |
|---|---|
| Entrée | 31 |
| Vegetables | 31 |
| Seafood | 22 |
| Sweets | 22 |
| Medicine/Health supplement | 20 |
| Fruit | 17 |
| Grains | 16 |
| Soft drink | 13 |
| Condiment | 10 |
| Meat/Eggs | 10 |
| Soybeans/Dried fruit | 6 |
| Dairy products | 2 |
| Total | 200 |

Table 1: Statistics of food types in FRDB.

parts: multiple-choice (MC) QA pairs and a knowledge database. The QA pairs contain 1,000 samples that cover 200 types of food. The categories of foods are shown in Table 1.

The possible answers to the question falls into three choices based on the increasing degree of recommendations ranging from 1 (avoid) to 3 (highly recommend). Each type of food has five recommendation rating corresponding to five groups: pre-pregnancy, pregnancy, postpartum, lactation, and infant. Additionally, we build a knowledge database that contains 7,588 entries; details of the entries are shown in Table 2. The distribution of sentence length in the knowledge database is shown in Figure 2.

| | Mean | Max | Min | Std. |
|---|---|---|---|---|
| # of words | 88 | 248 | 12 | 27 |

Table 2: Statistics of the entries FRDB knowledge database. **Std.** stands for standard deviation.

All the information has been verified by the health-domain professionals. During the verification, we remove the text that is ambiguous to the human annotators. Two samples of knowledge are shown in Table 3. Sample questions are available in the Appendix A.

## 4 Method

Typically, a retrieval-augmented LLM consists of three components (shown in Figure 1), a knowledge database, a retriever, and the LLM. The knowledge

| High quality knowledge | Ambiguous knowledge |
| --- | --- |
| Consuming mushrooms after childbirth is beneficial for postpartum recovery, constipation relief, and promoting lactation due to their rich B vitamins, protein, and amino acids. A moderate amount of intake based on the recovery status is recommended. | Postpartum mothers are safe to consume a moderate amount of cake. Cakes are easier to digest and absorb for postpartum mothers with weaker gastrointestinal systems. However, cakes have relatively smaller nutritional diversity and they should be consumed together with vegetables, fruits, and meats to make the nutrition more balanced. |

Table 3: Examples from the knowledge database. The ambiguous ones are excluded from our dataset.

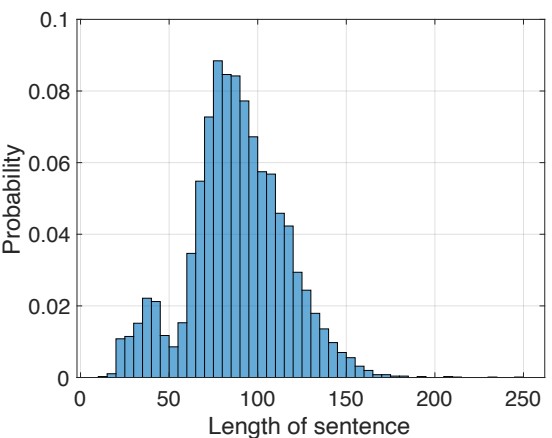

Figure 2: The distribution of sentence length in FRDB knowledge database.

database contains all available domain-specific knowledge. The retriever applies the question as a query to search for the relevant information from the knowledge database. The retrieved information is then formulated as the context packaged together with questions as a prompt for LLM to generate an answer. Our proposed methods are able to compress the retrieved information and formulate shorter context but maintain the effectiveness of retrieval augmentation. In this section, we go through the pipeline of a retrieval-augmented LLM system for QA and introduce our proposed token compression methods.

## 4.1 Information Retrieval

Generally, the first step for LLM's retrieval augmentation is knowledge retrieval. Given an user query $x$, the retriever extracts $k$ pieces of information from the knowledge database $\mathcal{D} = \{d_1, d_2, \cdots, d_m\}$ that are most likely to be relevant to $x$. There are two mainstream retrieval methods: dense retrieval (Karpukhin et al., 2020; Ni et al., 2021) and sparse retrieval (Robertson et al., 2009). Dense retrieval first encodes queries and documents into dense embeddings (Huang et al., 2013; Yi et al., 2019) using pre-trained neural en-

coders and then finds a query's nearest neighbors in the embedding space using a relevancy measure such as cosine similarity (Yu et al., 2021). Sparse retrieval, on the other hand, maps queries and documents into a high-dimensional space with methods such as TF-IDF (Sparck Jones, 1972; Jones, 1973) and the most relevant documents are returned to the user as the answer. Typical example of sparse retrieval is BM25 (Robertson et al., 1995).

Here we evaluate both dense and sparse retrieval methods. For dense retrieval, we follow a similar process from Huang et al. (2013): we first encode the text using the GPT-embedding (OpenAI, 2022) provided by OpenAI, then deploy vector database FAISS Index (Johnson et al., 2019) to store the embeddings, enabling faster manipulation on them. For sparse retrieval, we deploy BM25 (Robertson et al., 1995) which is considered the standard way.

### 4.1.1 Next Sentence Prediction

The retrieved top-$k$ results $\mathcal{D}_k = \{d_1^k, d_2^k, \cdots, d_k^k\}$ ($d_i^k$ are the $i$-th retrieved elements from the original set $\mathcal{D}$ where $1 \leq i \leq k$) are deemed as the most relevant ranked by the retrieval method. Using only the top-1 result as the context is not always reliable, but using more retrieved results will consume more space in the input tokens, leading to higher costs; therefore, to improve reliability without incurring additional costs, we propose to use next-sentence prediction (NSP) as a second-stage ranking method. It is based on an intuitive assumption that *the retrieved information is more likely to be predicted as the next sentence of the question if it is more related to the question*. The implementation of this approach is based on the pre-trained NSP module from BERT (Devlin et al., 2018); the selected sentence $s$ with maximum probability from NSP is selected as the best result (See Equation 1).

$$s = d_i^k$$
$$i = \underset{i}{\arg\max} \left\{ p(x, d_1^k), \cdots, p(x, d_k^k) \right\} \quad (1)$$

We conduct experiments to evaluate the impact of including NSP into the retrieval-augmented LLM. Here we use OpenAI's GPT-3.5-turbo as the base LLM and evaluate it on the FRDB dataset using the top-1 retrieval results as the context. The result is shown in Table 4. There is a minor performance gain using NSP with both GPT-embedding and BM25 and thus we keep this NSP module in all our later experiments.

| Method | Acc. (%) |
|---|---|
| Embedding | 89.1 |
| Embedding +NSP | 90.2 |
| BM25 | 83.4 |
| BM25+NSP | 84.9 |

Table 4: Performance comparison of retrieval methods with and without NSP. The retrieved sentences are directly used as context and fed into GPT-3.5-turbo. **Acc.** stands for accuracy.

From the experiment, we also see that the combination of dense retrieval with the NSP approach obtain the highest accuracy. We tune the value of $k$ by searching it from 1 to 10 and perform corresponding evaluation on the FRDB dataset. The experiment results are shown in Figure 3. We find that $k = 3$ is the optimal choice and we will adhere to this value in all our subsequent experiments.

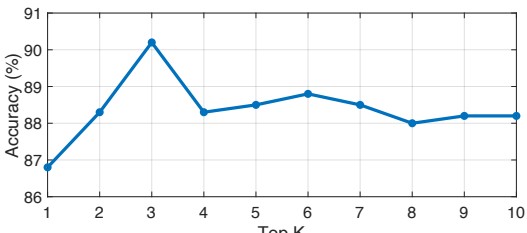

Figure 3: Evaluation of the retrieval performance using GPT-embedding and NSP with different choices of $k$.

## 4.2 Token Compression

The retrieval text is usually long and easily consume a huge amount of space from the input tokens during the API calls while using commercial LLMs. In order to mitigate this, we propose two methods to compress the retrieved text. The first one is the summarization compression which produces shorten the original text by rephrasing. The second method is semantic compression which we perturb the original sentence and rank the impact of the semantic change from each word in the sentence. The

words with lower semantic impact on the sentence are removed.

### 4.2.1 Summarization Compression

Summarization models like the mT5 model (Xue et al., 2020) have been widely used in many applications to shorten the input text, but they could not output summary with arbitrary length due to the constraint of its training data. To solve this, we propose to build a summarization model that is able to output summary with various lengths.

To build such a model, we leverage the power of self-instruct (Wang et al., 2022) where we use GPT-3.5-turbo to generate training datasets. The procedure of the data generation is shown in Figure 4. First, we start with a text $x$ from the dataset, then pack it with additional prompt instruction as we illustrated in Figure 4 and send it to GPT-3.5-turbo to generate a summary. If the length of the summary meets requirements, the procedure is ended; otherwise, a follow-up prompt will instruct GPT-3.5-turbo to further shorten the summary to the desired length. By doing this, we build a collection of training datasets with different summary length. We build three datasets that are 30%, 50%, and 70% of their original length. Each dataset is used to fine-tune one summary model independently. We randomly extract from FRDB and generate 400, 50, and 50 samples for training, validation, and testing respectively. Training on the generated datasets not only enables the model to produce summaries of the desired length, but also familiarizes the model with domain-specific information by doing further domain adaptation (Gururangan et al., 2020).

### 4.2.2 Semantic Compression

We propose another compression method based on perturbations of the original sentence and ranking the impact of the semantic importance for each word in the sentence where words with less importance will be removed. We deploy a multi-lingual sentence-transformer (Reimers and Gurevych, 2020) to encode a sentence into embedding $\chi_0$. Then we iteratively remove one word in the sentence and obtain an updated embedding $\chi_i$ where $i$ is the index of the word in the sentence and $n$ is the number of words in the sentence. We have a new set $\mathcal{L}$ that tracks the Euclidean distance between the original and perturbed embedding $\mathcal{L} = \{L2(\chi_0, \chi_1), \ldots, L2(\chi_0, \chi_n)\}$. We denote $p_j$ as the value of the $j$-th percentile in $\mathcal{L}$. The $\mathcal{L}_j$ is the new subset that removed the bottom

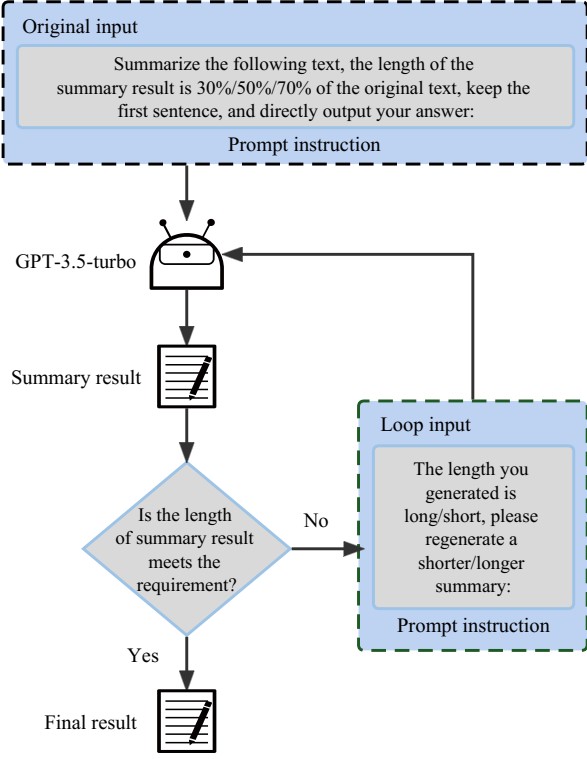

Figure 4: Self-instruct training data generation.

$j$-th percentile elements:

$$\mathcal{L}_j = \{\omega \in \mathcal{L}, \omega > p_j\} \qquad (2)$$

The words corresponding to the elements in set $\mathcal{L}_j$ are extracted as the context for the LLM.

## 5 Evaluation

### 5.1 Experiment Setup

We conduct studies on the FRDB dataset. The summarization compression is based on the pre-trained mT5-multilingual-XLSum (Xue et al., 2020). The maximum input and output length is set to 512, the learning rate is 2e-5, the number of epochs is set to 10, the batch size is 2, and the rest of the settings follow the default settings from the models. The training of the mT5 models are conducted on the server that contains an AMD EPYC 7763 CPU, 256GB RAM, and NVIDIA 4090 GPU with 24GB memory. Following the method described in section 4.2.1, three shortened versions of the summarization dataset with 30%, 50%, and 70% of its original length are generated. Each version is used to fine-tune an mT5 summarization model. The pre-trained multi-lingual sentence-transformer (Reimers and Gurevych, 2020) is used as an embedding encoder for semantic compression. Three compressed sentences in 30%, 50%,

and 70% of their original length are generated. The GPT-3.5-turbo (OpenAI, 2023c) is used for processing prompts generated by our methods.

### 5.2 Token Compression Results

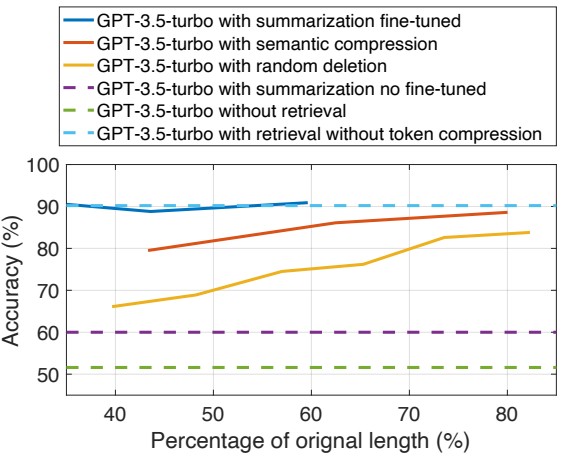

Figure 5: Performance comparison of token compression methods. The horizontal dashed lines represent accuracy from the methods that do not output variable token lengths.

We conduct experiments on FRDB which contains 1,000 maternity/infant food related domain-specific multiple-choice questions, each of which only contains one correct answer. Three sentence-compression methods are evaluated in our experiments: 1) random deletion which randomly deletes words from a sentence, 2) summarization compression, and 3) semantic compression.

The experiment results are shown in Figure 5. To construct the baseline, we evaluate the GPT-3.5-turbo performance in two configurations: without retrieval and with retrieval but without token compression. We observe from the results that, once additional information is fed into the model as context, the accuracy immediately improves, from 51% to 90.2%. This shows the benefit that retrieving domain information brings to the LLM.

Next, we compare using the original pre-trained mT5 based model with using the version where the model is fine-tuned on the datasets generated by self-instruction. With fine-tuning, the accuracy improves dramatically from 60% to 90.6%. The summarization model without fine-tuning output has an average length of 15 words, compare with 88, 46, and 21 words for our 70%, 50%, and 30% length dataset fine-tuned model respectively. It shows that the summarization model might remove critical information by mistake if the input text

is on a topic that the summarization model is not familiar with. A small fine-tuning set (400 samples) is enough to help the summarization model adapting to the new domain.

The second compression method we propose is semantic compression. It has better flexibility to generate variable lengths of tokens but delivers lower performance than the summarization compression method. The baseline for our experiment is random deletion where we randomly delete a certain percentage of words. This random deletion consistently scores lower performance than both of our proposed algorithms.

### 5.3 Cost Reduction Comparison

Our proposed summarization compression method is tested on a server with one NVIDIA 4090 GPU (24GB), 32GB RAM and Intel i7 CPU. The runtime for such a system is on average 0.383s per sample. A similar server on the AWS, i.e., g5.2xlarge, has an A10G GPU, 32GB RAM, and 8-core vCPU and the hourly price for such a system is \$0.485. In one hour, such a system can process approximately 9,400 summarizations, so the cost per summarization is \$5.16e-5. Assume the system utilization rate is only 50%, it means that the cost per summarization is \$1.0319e-04.

The GPT-3.5-turbo itself does not have enough knowledge to precisely answer the question from our dataset (51% accuracy if without additional context in the prompt). Thus, both the common retrieval-augmented GPT-3.5-turbo and our system (retrieval-augmented GPT-3.5-turbo with additional token compression) require dense retrieval to reach acceptable performance, and the retrieval cost, which is \$0.0001 per 1,000 query tokens, is the same for both systems. Since the original questions are typically short, we can assume that the average length of them is about 128 tokens, which translates into \$1.2500e-05 per question for the retrieval. Assuming at full size the input has 512 tokens and the output has 64 tokens, the total cost for the common retrieval-augmented GPT-3.5-turbo (for both retrieval and QA using API calls) is about \$8.9050e-04 per question. In comparison, our algorithm can compress the retrieved context of GPT-3.5-turbo to 35% of its original length, which translates into an averagely 50% of reduction during the API calls. Thus, The cost using our token compression system is around \$6.1869e-04 per question. It reduces the overall costs by 30%.

### 5.4 Information Entropy vs. Accuracy

In the next experiment, we investigate the impact of the information entropy on the accuracy of different token compression methods. We measure the word frequency from the FRDB dataset and use it as a probabilistic model to calculate the information entropy of each word. The mean word information entropy is calculated on each sentence to normalize the entropy. The results for the three token compression methods is shown in Figure 6. The random deletion method removes words randomly which leads to the average information entropy for different sentence lengths approximately the same. On the other hand, the semantic compression algorithm removes the words that have less semantic meaning. Our experiment shows that, the average information entropy goes lower as sentences become shorter, indicating that the sentence becomes less compressible. Additionally, the average word information entropy is positively correlated with the accuracy when semantic compression is used, showing that higher information will benefit the model performance. On the contrary, the summarization compression shows distinct phenomenon. Instead of naively removing words, the summarization compression compresses the original sentences into different lengths by rephrasing sentences. By doing this, the shortened sentences obtain lower average information entropy but the accuracy stays at a similar level in comparison with the original sentences. The lower average information entropy indicates that sentences become more condensed but the semantic of the sentences stays approximately the same.

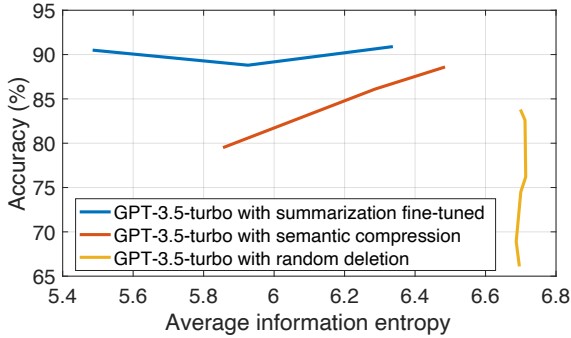

Figure 6: Impact of average information entropy on accuracy.

Next, we investigate the impact of cosine similarity between original and compressed sentences on accuracy and we find a positive correlation between accuracy and cosine similarity value. It indicates

closer to the original semantic meaning would produce better accuracy.

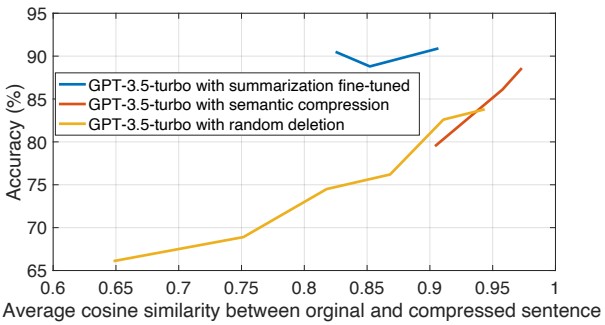

Figure 7: Impact of cosine similarity between original and compressed sentence on accuracy.

## 5.5 Summarization Statistics

Our summarization model is built based on the pre-trained mT5 model and fine-tuned on our self-instruct generated dataset. We generate three datasets which are 30%, 50%, and 70% of the length compared to their original text. Three different models are fine-tuned independently. Figure 8 shows the distribution of sentence length from our three fine-tuned models. At 70% compression, the summary text shifts from an average of 88 words to 46 words for 50% length and 21 words for 30% length.

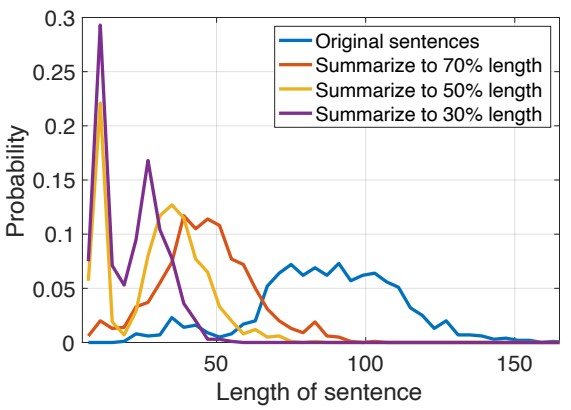

Figure 8: Distribution of sentence lengths with different summarization lengths.

## 5.6 Compression Rate vs. Entropy

We conduct a study on the compression rate of our mT5 model fine-tuned with the 30% length dataset. Our input consists of 1) the original length of the sentence, 2) average word entropy, and 3) accumulated word entropy. We deploy a simple linear regression algorithm to predict the compression

rate. The result is shown in Fig. 9. We find that, there is a positive correlation (0.31) between our selected input and compression rate with an RMSE of 11.4% and R-squared of 9.6%. This indicates the compression rate of each sentence can be estimated prior to the summarization process.

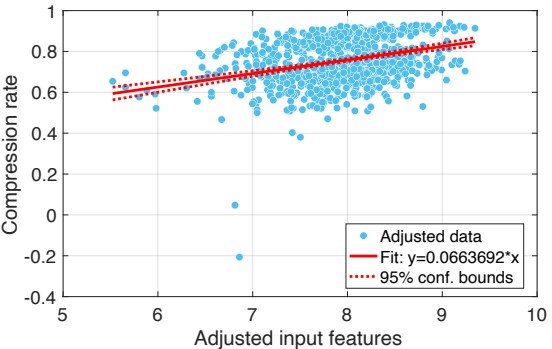

Figure 9: Visualization of multi-variable linear regression on predicting compression rate.

## 5.7 Ablation Study

From our experiments, we find the summarization compression method delivers the best performance. Here we compare different retrieval methods and investigate what the optimal settings are. Four configurations are evaluated: embedding only, BM25 only, embedding first then BM25, and BM25 first then embedding. The first two configurations are straightforward; in the third configuration, we apply the embedding-based method to extract the top-$q$ results, then apply BM25 to extract top-$k$ results from the $q$ results where $q \geq k$. In the fourth configuration, we reverse the order where we first extract the top-$q$ results using BM25 and then extract the top-$k$ results from the $q$ results using embedding-based methods. We set $k = 3$ based on previous experiments. The evaluation results are shown in Table 5. We find the straightforward dense retrieval approach achieves the best performance.

## 6 Conclusion

In this paper, we propose two methods that reduce the token size for retrieval-augmented LLM. Additionally, we propose a food recommendation dataset contains domain-specific knowledge to benchmark the performance of retrieval-augmented LLM performance on the GPT-3.5-turbo. We carefully select a subset that focuses on 200 types of food recommendations for maternity and infant people. Without retrieval augmentation, the commercial GPT-3.5-turbo model is only able to get

| Method | Top-$q$ | Acc. (%) |
|---|---|---|
| Embedding | n/a | 90.9 |
| Embedding+BM25 | 10 | 89.7 |
| Embedding+BM25 | 100 | 88.0 |
| BM25 | n/a | 74.7 |
| BM25+Embedding | 10 | 89.3 |
| BM25+Embedding | 100 | 89.8 |

Table 5: Evaluation of different retrieval algorithm configurations. The top-$q$ column indicates the $m$ value used in the search. **Acc.** stands for accuracy.

51% percent of the question right, compared to 90.2% with retrieved context. We use this 90.2% as our goal and compare the performance of different token compression algorithms. Our summarization-based approach achieves the best performance, reaching 90.5% of accuracy with 65% token reduction on the retrieved context. It indicates that the summarized text maintains a similar level of critical information but with a significantly shorter length, showing promising ability of our proposed method. The semantic compression method can further remove the words that have lower semantic meaning and provides a more flexible way to trade-off the length of sentences with accuracy.

## Limitations

The goal of our model is to reduce the token size for retrieval-augmented LLMs. The compression rate is determined based on the methods' ability to condense sentences while preserving as much of their essential information as possible. If the sentences are already short and compacted in meaning, the compression rate won't be able to be low if we want to maintain most of the critical information within the sentence. Our algorithm is designed for large commercial LLMs and smaller open-source LLMs may not experience similar levels of performance gain.

## Ethics Statement

Our proposed methods are designed solely for the goal of reducing cost while maintaining the performance using commercial LLMs through API calls. The algorithms are not designed for activities that are in violation of human rights or with military purposes. The data we collected for our proposed dataset does not contain any privacy information.

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

## Appendix

### A Example question for FRDB dataset

Is it suitable for an infant to consume/have hot chocolate? (1) Recommend, (2) Neutral, (3) Avoid

1. Food is beneficial to the current stage, and excessive consumption will not cause physical abnormalities.

2. Advocate a healthy lifestyle and advise users to intake relatively limited amount of foods, including foods with excessive salt, oil, sugar, and so on.

3. There is evidence that it will cause harm after eating; Authoritative literature points out that eating is forbidden at a certain stage.

### B Sentence compression examples

Examples are shown in Table 6.

### C QA examples

Examples of QA are shown in Table 7.

### D Knowledge examples

Examples of knowledge utilized for answering the questions are shown in Table 8.

| Dataset | Original | Compressed |
|---|---|---|
| FRDB | Babies are strictly prohibited from consuming donkey meat buns. The reason is that donkey meat buns usually contain a large amount of spices and seasonings, with high salt content, which will increase the metabolic burden on the baby's kidneys. Furthermore, infants and young children are in a critical period for developing taste preferences, and frequently consuming donkey meat buns may affect the formation of their taste buds. It is not recommended to feed babies donkey meat buns. | In the first-instance procedure of a public prosecution case, if the participants or observers disrupt the courtroom order, the presiding judge shall handle the situation according to the law. Unauthorized recording, videotaping, photographing, or disseminating trial proceedings through mail, blog, or microblog may lead to the temporary seizure of storage media or relevant equipment. |

Table 6: Token compression examples. Both original and compressed sentences are translated into English from Chinese.

| Questions | Questions (English translation) | Answer |
|---|---|---|
| 对于'根据 "产妇少吃鹅肝。原因是：产后妈妈适量食用鹅肝有利于促进伤口的愈合，弥补生产失血和重建肝脏铁储备，提高乳汁质量。还能提高妈妈的免疫力，具有抗氧化、缓解衰老的作用。建议产后妈妈可适量食用。" 产妇食用/饮用鹅肝'，下列哪个选项最适合用于描述上述行为？选项：(1) 推荐，(2) 中立，(3) 避免 | According to "Maternity should eat less foie gras. The reason is: moderate consumption of foie gras after childbirth can promote wound healing, make up for blood loss during childbirth, rebuild liver iron reserves, improve breast milk quality. It can also improve the immunity of mothers, and has an antioxidant and anti-aging effect. It is recommended that postpartum mothers can consume it in moderation." Which of the following options is most suitable to describe the behavior of postpartum mothers consuming/eating foie gras? Options: (1) Recommend, (2) Neutral, (3) Avoid | Neutral |
| 对于'根据 "备孕女性可以吃咖喱。原因是：咖喱属于混合调制的香料，能调节肠胃蠕动，提高食欲，其辛辣程度根据配料而变，过于辛辣的咖喱对胃有一定的刺激性，备孕期女性可以根据自己的口味喜好选择合适辣度的咖喱。" 备孕女性食用/饮用咖喱'，下列哪个选项最适合用于描述上述行为？选项：(1) 推荐，(2) 中立，(3) 避免 | Regarding "Females preparing for pregnancy can eat curry. The reason is that curry is a mixed seasoning that can regulate intestinal movement, increase appetite, and its spiciness varies according to the ingredients. Curry that is too spicy can stimulate the stomach to some extent. Females preparing for pregnancy can choose curry with appropriate spiciness based on their own taste preferences." Which of the following options is most suitable to describe the behavior of females preparing for pregnancy consuming/eating curry? Options: (1) Recommend, (2) Neutral, (3) Avoid | Recommend |
| 对于'根据 "6月大的宝宝少吃玉米汁。原因是：玉米汁富含维生素、矿物质和碳水化合物，可为宝宝提供能量，促进其生长发育，且吸收率较高。6月龄以后的宝宝可少量食用，注意鲜榨玉米汁不要添加糖，以防摄入过多的糖，不利于宝宝的口腔健康。" 6月龄的宝宝食用/饮用玉米汁'，下列哪个选项最适合用于描述上述行为？选项：(1) 推荐，(2) 中立，(3) 避免 | Regarding "Babies at the age of 6 months should consume less corn juice. The reason is that corn juice is rich in vitamins, minerals and carbohydrates, which can provide energy for babies, promote their growth and development, and has a high absorption rate. Babies over 6 months old can consume it in moderation, but be mindful that freshly squeezed corn juice should not contain added sugar to prevent excessive intake of sugar, which could be detrimental to baby's oral health." Which of the following options is most suitable to describe the behavior of a 6-month-old baby consuming/drinking corn juice? Options: (1) Recommend, (2) Neutral, (3) Avoid | Neutral |

Table 7: QA examples. For all examples, the answers are chosen from 3 options, (1) Recommend, (2) Neutral, and (3) Avoid.

| Knowledge | Knowledge (English translation) |
|---|---|
| 备孕女性可以吃土豆泥。原因是：备孕人群可以食用，不过土豆泥升糖较快，建议一次不要吃太多。 | Females preparing for pregnancy can eat mashed potatoes. The reason is that it is safe for this group to consume mashed potatoes. However, mashed potatoes have a high glycemic index, so it is recommended not to eat too much at once. |
| 宝宝不能吃生鱼片。原因是：鱼肉中含有大量的不饱和脂肪酸（尤其是DHA），有助于促进宝宝大脑及视力发育。但生鱼片如果处理不当，容易感染病菌和寄生虫。不建议给宝宝吃生鱼片。 | Babies should not eat raw fish slices. The reason is that fish contains a large amount of unsaturated fatty acids (especially DHA), which can help promote the development of the baby's brain and vision. However, if raw fish slices are not properly processed, they can easily become contaminated with bacteria and parasites, posing health risks to babies. Therefore, it is not recommended to feed raw fish slices to babies. |
| 孕妇可以吃罗非鱼。原因是：罗非鱼中含有非常丰富的不饱和脂肪酸、蛋白质和多种氨基酸，容易被人体消化吸收。能为孕妈提供多种营养物质，帮助胎儿骨骼生长和神经系统发育。 | Pregnant women can eat tilapia. The reason is that tilapia is rich in unsaturated fatty acids, protein and various amino acids, which are easily digested and absorbed by the human body. It can provide pregnant women with various nutrients to help promote fetal bone growth and development of the nervous system. |

Table 8: Examples of knowledge that are utilized for answering the questions.