# OpenReview forum: "TCRA-LLM: Token Compression Retrieval Augmented Large Language Model for Inference Cost Reduction"
_EMNLP/2023/Conference — EMNLP 2023 Findings_

### Official Review · Reviewer_VUd2 · 2023-08-04

**Soundness:** 4

**Ethical Concerns:**

Yes

**Excitement:**

3: Ambivalent: It has merits (e.g., it reports state-of-the-art results, the idea is nice), but there are key weaknesses (e.g., it describes incremental work), and it can significantly benefit from another round of revision. However, I won't object to accepting it if my co-reviewers champion it.

**Justification For Ethical Concerns:**

Further clarification about the collection process of FRDB dataset is required to check the ethical concern.

**Paper Topic And Main Contributions:**

In this paper, they proposed two approaches (summarization and semantic compression) to compress the token size for retrieval augmented LLMs. They also proposed a food recommendation dataset which includes the domain of recomending foods that are safe to consume for women before/during/after their pregnancy or infant. By applying these two techniques, they reached a significant amount of token compression while maintaing the performance (e.g. 65% token reduction while havinf 0.3% performance improvement).

**Questions For The Authors:**

QA: Could you please elaborate more on the generalization of the model? (refer to first concern on previous section)

**Reasons To Accept:**

1. A novel combination of token compression methods, including summarization and semantic compression to reduce the input tokens of LLMs.
2. Proposing a new dataset for food recommendation which includes the domain of recomending foods that are safe to consume for women before/during/after their pregnancy or infant.
3. Achieving significant token compression while maintaining the performance.

**Reasons To Reject:**

1. A weak novelty for a long paper. They just combined two already proposed methods for token compression, and the main improvement is comming from fine-tuning summarizer.
2. The generalization of the approach to other domains is not straight-forward, as the modules should be fine-tuned on the new domain.

**Reproducibility:**

5: Could easily reproduce the results.

**Reviewer Confidence:**

4: Quite sure. I tried to check the important points carefully. It's unlikely, though conceivable, that I missed something that should affect my ratings.

---

> ### Author Rebuttal · Authors · 2023-08-27
>
> 1.	A weak novelty for a long paper. They just combined two already proposed methods for token compression, and the main improvement is comming from fine-tuning summarizer.The generalization of the approach to other domains is not straight-forward, as the modules should be fine-tuned on the new domain.
>
> Thanks for your comments. We acknowledge our paper is an application paper that focuses on engineering a system to solve the problem that appears in the business deployment. During our daily business operation, we are experiencing high operating costs where every penny we save will translate into a penny of profit. During the construction of the system, we put a lot of engineering thought into it, where a higher system complexity will require higher server costs which may eat away the potential savings. By balancing the server cost and algorithm performance, we found an easy yet efficient method to reduce the cost by 30% which is significant for the survival of our business. Creating a generalizable solution for the business environment is always not easy, we need to balance the cost and performance of the system. From our experiment, a model fine-tuned for each domain is optimal to balance the costs and performance.

---

### Official Review · Reviewer_9Fyp · 2023-08-12

**Soundness:** 4

**Excitement:**

4: Strong: This paper deepens the understanding of some phenomenon or lowers the barriers to an existing research direction.

**Paper Topic And Main Contributions:**

API calls to commercial large language models are very costly. Data augmentation is used provide additional context to these calls. This increases the cost as it increases the number of tokens in the call.
To reduce the cost, author’s propose token compression scheme for such retrieval augmented large language model. They call it Token Compression Retrieval Augmented Large Language Model (TCRA-LLM).
Author’s present 2 approaches to compress the additional relevant information retrieved from the knowledge database.
1.	Summarization – generate summaries of different lengths. Summarization model is trained using the data generated by self-instruct model.
2.	Semantic compression - words with less semantic impact  (such as filler words) are removed based on the Euclidian distance between the original and modified sentence embeddings.
To improve reliability of additional context from knowledge databases, author’s presented next-sentence prediction (NSP) to rank the retrieved information.

**Reasons To Accept:**

1. Relevant solution for today’s LLM applications.
2. The solution is explained well.
3. Good set of experiments were used to demonstrate the benefits of suggested approaches.
4. Overall cost analysis, difference in before/after will be added in final version.
5. Increase in overall inference time will be added in final version.
6. Considering that it’s a classification task, other performance metrics will be added in final version.

**Reasons To Reject:**

1. The increase in time and effort for data creation and summarization model training needs to studied.
2. Generality for different domains and dataset sizes should be studied.

**Reproducibility:**

3: Could reproduce the results with some difficulty. The settings of parameters are underspecified or subjectively determined; the training/evaluation data are not widely available.

**Reviewer Confidence:**

4: Quite sure. I tried to check the important points carefully. It's unlikely, though conceivable, that I missed something that should affect my ratings.

---

> ### Author Rebuttal · Authors · 2023-08-27
>
> 1.	Considering that it’s a classification task, other performance metrics need to be studied. Increase in overall inference time needs to be studied. Overall cost and effort impact need to studied.
>
> Thanks for your comments. We have conducted calculations on additional performance metrics.
>
> Acc: 0.909
> Weighted F1 Score: 0.908442161770658
> Macro F1 Score: 0.8865993509859474
> Micro F1 Score: 0.909
> Macro Recall Score: 0.8641033143866387
> Micro Recall Score: 0.909
> Weighted Recall Score: 0.909
> Macro Precision Score: 0.9195618277905991
> Micro Precision Score: 0.909
> Weighted Precision Score: 0.9130815891627097
>
> We will add an additional cost study in the revised draft. The other reviewer also asked the study in terms of costs. In terms of inference time, the difference between the use of retrieval augment with and without the token compression is about 0.383s. The other part is the same. The average time for embedding creating using OpenAI API is 0.468s, and the average time for gpt3.5 is 0.813s.
>
> Let me do a cost study between renting an AWS server and using the GPT3.5 service.
>
> Our mT5 system was tested on a server with one Nvidia 4090 GPU (24GB), 32GB RAM and Intel i7 CPU. The runtime for such a system is on average 0.383s per sample.  On the AWS the similar g5.2xlarge server has A10G GPU, 32GB RAM, and 8-core vCPU. The hourly price for such a system is $0.485. In one hour, such a system can process about 9,400 summarizations, so the cost per summarization is $5.16e-5. Let’s assume the system utilization rate is only 50%, which means per summarization is $1.0319e-04.
>
> The GPT3.5 does not have the knowledge to answer the question from our dataset (51% accuracy without supplying the context in the prompt). So, both our system and the gpt3.5 require embedding-based retrieval, so such cost is the same for both systems which is $0.0001/1k tokens. The questions are usually short and we assume the length is about 128 tokens, which translates into $1.2500e-05 per question.
>
> Assuming at full size the input is 512 tokens and the output is 64 tokens. We can calculate each total QA cost (embedding+QA) about 0.0015*0.5+0.002*0.064+1.2500e-05 =8.9050e-04 per question.
>
> Our algorithm can compress the context to 35% of its original length, which translates into an overall 50% token saving on average. So, the new costs for answer-question using our token compression system are 0.0015*0.25+0.002*0.064+1.0319e-04+1.2500e-05 = $6.1869e-04. It reduces the overall QA costs (embedding+QA) by 30%.

---

### Official Review · Reviewer_Z9sQ · 2023-08-14

**Soundness:** 3

**Excitement:**

4: Strong: This paper deepens the understanding of some phenomenon or lowers the barriers to an existing research direction.

**Missing References:**

Exploiting Context in LLMs:
Di Liello, L., Garg, S., & Moschitti, A. (2023). Context-Aware Transformer Pre-Training for Answer Sentence Selection. (https://arxiv.org/abs/2305.15358)

**Paper Topic And Main Contributions:**

In this study, the researchers investigate the efficacy of Language Models (LLMs) in two scenarios: (i) when enhanced with supplementary context to improve responses, and (ii) when input context is shortened using various techniques, such as summarization and the removal of low informative tokens.

The results are promising and the described techniques provide important gains in accuracy. However, the paper lacks details about several
experiments and the performance are evaluated only with a small and domain specific dataset.

**Update after rebuttal**

The authors addressed some of my concerns, however:
- a lot of important details and hyper-parameters are still hidden in the supplementary materials, while they should be included in the text or appendix of the paper;
- the comparison between the cost of using the proposed solution with compressed context and using only ChatGPT should be included at least in the appendix of the paper;
- the evaluation is still performed on a single small dataset, reducing the possibility of applying the proposed technique to other domains;
- the paper is still full of typos and hard to follow

I've updated the scores accordingly, increasing the overall excitement because the topic and the solutions addresses by this paper are interesting. I kept a soundness of 3 for the reasons explained above.

**Reasons To Accept:**

- The paper provides an interesting study to add context to LLMs for better responses and methodologies to reduce input token length while maintaining a comparable level of performance;
- The results show that context really helps in identifying the correct answer to a question, improving accuracy.

**Reasons To Reject:**

- The process of data collection and annotation for the proposed dataset (FRDB) lack details;
- The experiments are missing details, such as how the FRDB dataset was divided in train/dev/test splits;
- The authors claim they use mT5 to compress context and spend less money on OpenAI API. However, what is the cost of summarizing context with mT5? A comparison is needed. The same applies to the embedding-based retriever, which uses again OpenAI embedding APIs;
- The generation hyper-parameters used for mT5 are missing, such as greedy search, beam search, ...
- The authors did not show which prompt they used to add context to OpenAI GPT, reducing reproducibility;
- The evaluation is performed on a single and very small dataset in the health domain, thus reducing the possibility to apply the proposed techniques to other topics;
- In Figure 6, is it not clear how a Average information entropy of ~6.5 can lead to very different accuracies at the same time.
- Some claims are not explained. For example, on line 288, the authors say that "LLM could only output fixed-length summary due to the fixed length of its training data.". What about length penalties and beam search to select a summaries that better fit the given requirements?

**Reproducibility:**

3: Could reproduce the results with some difficulty. The settings of parameters are underspecified or subjectively determined; the training/evaluation data are not widely available.

**Reviewer Confidence:**

4: Quite sure. I tried to check the important points carefully. It's unlikely, though conceivable, that I missed something that should affect my ratings.

**Typos Grammar Style And Presentation Improvements:**

- The paper is not well written and not easy to follow;
- Some sentences are not complete, for example on line 175 or 508;
- The paper contains many grammar typos.

---

> ### Author Rebuttal · Authors · 2023-08-27
>
> Thanks for your comments. I would like to address the comments below.
>
> 1.	The process of data collection and annotation for the proposed dataset (FRDB) lacks details;
>
> This FRDB dataset is collected from a combination of frequently asked questions from social platforms and verified by our nutrition experts. We will add this additional detail in the paper. The full 1000 question answer FRDB dataset is available in the supplementary material. The retrieval knowledge database that contains 7,588 entries is subject to our internal approval. We will release it once it is approved.
>
> 2.	The experiments are missing details, such as how the FRDB dataset was divided into train/dev/test splits;
> The question-answer part of FRDB is included in the supplementary. We are doing retrieval augmented generation leveraging the power of ChatGPT. So, all the QA pairs are test sets.
>
> 3.	The authors claim they use mT5 to compress context and spend less money on OpenAI API. However, what is the cost of summarizing context with mT5? A comparison is needed. The same applies to the embedding-based retriever, which uses again OpenAI embedding APIs;
> Let me do a cost study between renting an AWS server and using the GPT3.5 service.
>
> Our mT5 system was tested on a server with one Nvidia 4090 GPU (24GB), 32GB RAM and Intel i7 CPU. The runtime for such a system is on average 0.383s per sample.  On the AWS the similar g5.2xlarge server has A10G GPU, 32GB RAM, and 8-core vCPU. The hourly price for such a system is $0.485. In one hour, such a system can process about 9,400 summarizations, so the cost per summarization is $5.16e-5. Let’s assume the system utilization rate is only 50%, which means per summarization is $1.0319e-04.
>
> The GPT3.5 does not have the knowledge to answer the question from our dataset (51% accuracy without supplying the context in the prompt). So, both our system and the gpt3.5 require embedding-based retrieval, so such cost is the same for both systems which is $0.0001/1k tokens. The questions are typically short and we assume the length is about 128 tokens, which translates into $1.2500e-05 per question.
>
> Assuming at full size the input is 512 tokens and the output is 64 tokens. We can calculate each total QA cost (embedding+QA) about 0.0015*0.5+0.002*0.064+1.2500e-05 =8.9050e-04 per question.
>
> Our algorithm can compress the context to 35% of its original length, which translates into an overall 50% of token savings on average. So, the new costs for answer-questions using our token compression system are 0.0015*0.25+0.002*0.064+1.0319e-04+1.2500e-05 = $6.1869e-04. It reduces the overall QA costs (embedding+QA) by 30%.
>
>
> 4.	The generation hyper-parameters used for mT5 are missing, such as greedy search, beam search, ...
>
> Due to the page length limit, the hyper-parameters were not included in the main text. It is available in our supplementary material. We have attached a fully functioning training, and inference code with all the parameters. You are also welcome to try out the system.
>
> 5.	The authors did not show which prompt they used to add context to OpenAI GPT, reducing reproducibility;
>
> We have attached some prompt examples in the supplementary material Tables 6, and 7. Also, I noticed, that we missed the multiple-choice option part in the prompt sample. I will revise this in the supplementary material.
>
> 6.	The evaluation is performed on a single and very small dataset in the health domain, thus reducing the possibility to apply the proposed techniques to other topics;
>
> Thanks for your suggestion. Our goal is to test out a dataset that is never trained in GPT3.5’s system, due to the vast large training set it has, this kind of dataset is hard to find. So, we picked this domain specific dataset on food recommendation. We agree that more experiments should be running to verify other domains also can benefit from our algorithm. We will provide some additional experiments in the camera-ready version.
>
> 7.	In Figure 6, is it not clear how a Average information entropy of ~6.5 can lead to very different accuracies at the same time.
>
> This phenomenon appears in the random deletion of words in the context. From our experiment, when we randomly delete words, the average information entropy per word does not change much but the semantic meaning of the sentence changed significantly. This experiment shows semantic meaning is the leading factor in impact on the accuracy.
>
> 8.	Some claims are not explained. For example, on line 288, the authors say that "LLM could only output fixed-length summary due to the fixed length of its training data.". What about length penalties and beam search to select summaries that better fit the given requirements?
>
> Our goal is to study the impact of compressed tokens on the QA tasks. In our experiment, we tried three different compression rates which compressed the sentence to 30%, 50%, and 70% of its original length. To achieve such a control experiment, we utilized the power of self-instruct learning from GPT3.5 to construct a dataset to fine-tune the summarization model with sentences compressed to 35%, 43%, and 60% of the original length. Length penalties and beam search are alternatives for controlling the length, but our experiment shows the summarization model fine-tuned on a small set of self-instruct generated data significantly outperformed the version without fine-tuning. In our case, the fine-tuning process is merely a necessity, and after fine-tuning we obtained a shorter length of summary which is our goal. So we did not have a strong motivation to further control the output length of the fine-tuned model.

---

### Meta-Review · Area_Chair_TGw3 · 2023-09-19

**Recommendation:** 3

**Metareview:**

This paper addresses a task of cost reduction when creating application using LLM APIs (for example, from OpenAI). This is indeed a bottleneck for many applications: while LLMs achieve impressive results, encoding user-generated information as context can get costly. In this sense the study provides an important solution to today's problems (see reviewers 1 and 2). The solution itself is based on existing techniques (cf. reviewer 3), however, the application is novel and impactful. Altogether, this study might be more appropriate for the industrial/applications track.

The experiments are convincing, showing a good compression rate with virtually no performance loss.

However, there are several shortcomings:

1) a more detailed cost analysis (see reviewer 1). this is important as the cost is the main focus here. this issue has been addressed thoroughly in the rebuttal -- and has to be incorporated into the next version
2) presentation. a lot of information (e.g. hyperparameters etc) is presented in the supplementary material and should be upgraded to the appendix; the paper needs proofreading. this is a minor issue that can be fixed for the next version
3) the experiments are run on a single domain. it is not clear how well the proposed solution generalize to other domains/settings (after all, the main motivation behind this study is to reduce cost for user domains, so the paper should focus on diverse use cases). This is a major issue raised by all the reviewers.

---

### Decision · Program_Chairs · 2023-10-07

**Decision:**

Accept-Findings

**Comment:**

This paper addresses a task of cost reduction when creating application using LLM APIs (for example, from OpenAI). This is indeed a bottleneck for many applications: while LLMs achieve impressive results, encoding user-generated information as context can get costly. In this sense the study provides an important solution to today's problems (see reviewers 1 and 2). The solution itself is based on existing techniques (cf. reviewer 3), however, the application is novel and impactful. Altogether, this study might be more appropriate for the industrial/applications track.

The experiments are convincing, showing a good compression rate with virtually no performance loss.

However, there are several shortcomings:

1) a more detailed cost analysis (see reviewer 1). this is important as the cost is the main focus here. this issue has been addressed thoroughly in the rebuttal -- and has to be incorporated into the next version
2) presentation. a lot of information (e.g. hyperparameters etc) is presented in the supplementary material and should be upgraded to the appendix; the paper needs proofreading. this is a minor issue that can be fixed for the next version
3) the experiments are run on a single domain. it is not clear how well the proposed solution generalize to other domains/settings (after all, the main motivation behind this study is to reduce cost for user domains, so the paper should focus on diverse use cases). This is a major issue raised by all the reviewers.